# *Candida tropicalis* Affects *Candida albicans* Virulence by Limiting Its Capacity to Adhere to the Host Intestinal Surface, Leading to Decreased Susceptibility to Colitis in Mice

**DOI:** 10.3390/jof10040245

**Published:** 2024-03-25

**Authors:** Kyle Roberts, Abdullah Osme, Carlo De Salvo, Eleonora Zoli, Janet Herrada, Thomas S. McCormick, Mahmoud Ghannoum, Fabio Cominelli, Luca Di Martino

**Affiliations:** 1Center for Medical Mycology and Integrated Microbiome Core, Department of Dermatology, University Hospitals Cleveland Medical Center, Case Western Reserve University, Cleveland, OH 44106, USA; kdr43@case.edu (K.R.); jlh269@case.edu (J.H.); tsm4@case.edu (T.S.M.); mag3@case.edu (M.G.); 2Department of Anatomic Pathology, University of Alabama at Birmingham, Birmingham, AL 35294, USA; aosme@uabmc.edu; 3Department of Pathology, School of Medicine, Case Western Reserve University, Cleveland, OH 44106, USA; cxd198@case.edu (C.D.S.); fabio.cominelli@uhhospitals.org (F.C.); 4Case Digestive Health Research Institute, School of Medicine, Case Western Reserve University, Cleveland, OH 44106, USA; exz174@case.edu; 5Department of Medicine, School of Medicine, Case Western Reserve University, Cleveland, OH 44106, USA

**Keywords:** *C. albicans*, colitis, *C. tropicalis*, mycobiome, biofilm

## Abstract

*Candida* (*C.*) infections represent a serious health risk for people affected by inflammatory bowel disease. An important fungal virulence factor is the capacity of the fungus to form *biofilms* on the colonized surface of the host. This research study aimed to determine the effect of a *C. tropicalis* and *C. albicans* co-infection on dextran sodium sulfate (DSS)-induced colitis in mice. The colitis severity was evaluated using histology and a colonoscopy. The mice were mono-inoculated with *C. albicans* or *C. tropicalis* or co-challenged with both species. The mice were administered 3% DSS to induce acute colitis. The biofilm activity was assessed using (2-methoxy-4-nitro-5-sulfophenyl)-5-[(phenylamino)carbonyl] 2H-tetrazoliumhydroxide (XTT) and dry-weight assays. The abundance of *C. albicans* in the colon tissues was assessed by immunohistochemistry. The co-challenged mice showed a decreased colitis severity compared to the mono-inoculated mice. The dry-weight assay demonstrated a marked decrease in *C. albicans* biofilm production in a *C. albicans* culture incubated with *C. tropicalis* supernatant. Immunohistochemical staining showed that *C. albicans* was more abundant in the mucosa of *C. albicans* mono-inoculated mice compared to the co-inoculated group. These data indicate an antagonistic microbial interaction between the two *Candida* species, where *C. tropicalis* may produce molecules capable of limiting the ability of *C. albicans* to adhere to the host intestinal surface, leading to a reduction in biofilm formation.

## 1. Introduction

Ulcerative colitis (UC) and Crohn’s disease (CD) are the two main forms of chronic inflammatory bowel diseases (IBDs). CD can affect any part of the gastrointestinal system, and patients can present with multiple symptoms, including abdominal pain, diarrhea, nausea, fatigue, cramping, and blood in the stools [1,2].

CD is a chronic disease that cannot be fully cured; the main clinical therapies are focused on slowing down CD progression using anti-inflammatory approaches such as biological therapies and steroids [3,4].

Multiple factors contribute to the pathophysiology of IBD and the consequent manifestation of a wide range of types and severities of symptoms, including the intestinal microbial communities (microbiome), immune responses, psychological stress, and the patient’s genetic susceptibility to the disease [5,6,7]. Specifically, genetically predisposed subjects can commonly experience an inappropriate immune response to intestinal commensal microbes [8].

Recently, several studies have shown that the fungal community (mycobiome), an essential and integral component of the intestinal microbial population, can affect the pathogenesis of CD [9]. The mycobiome can reside in any part of the digestive tract and mostly consists of commensal fungi. However, in some instances, intestinal fungal commensals can overgrow and become opportunistic pathogens, therefore contributing to the etiology of IBD, particularly in more susceptible individuals such as immunocompromised patients [10].

Clinical studies on immunocompromised individuals have shown that most fungal infections in CD patients are caused by *Candida* (*C.*) *albicans* and *C. tropicalis*, with these fungi identified as the most common pathogenic yeasts worldwide [11,12]. *C. albicans* and *C. tropicalis* are normal components of the human microbiome and are commonly present in the gastrointestinal system, epidermis, and genital tract [13].

*C. tropicalis* is characterized by a high resistance to antifungal treatments, such as amphotericin B and azole derivatives [14], and has been identified as the second most common pathogenic yeast in IBD patients, after *C. albicans*. It is significantly more abundant in CD patients compared to their non-CD relatives [15].

Candida species exist in both yeast and hyphal forms based on the surrounding environment, and they are the most commonly reported fungal species causing infections in IBD patients, particularly in the gastrointestinal tract [16]. One of the most important C. albicans virulence factors is its capacity to form polymicrobial biofilms (PMBs) on the colonized surface of the host, thereby promoting associations with several types of bacteria [17]. The ability of C. albicans to form polymicrobial associations indicates that crosstalk between the mycobiome and microbiome may negatively affect the host. The underlying mechanisms for this detrimental effect are attributed to the ability of yeasts to form filaments and secrete extracellular enzymes (aspartic proteinase and phospholipases) [18,19], leading to apoptosis, oxidative damage, and a significantly increased production of proinflammatory cytokines. This eventually induces an abnormal host inflammatory response [20], resulting in the breakdown of the epithelial cell lining and a leaky gut.

Since *Candida*-induced dysbiosis has been shown to be detrimental in both CD patients and CD mouse models [21], understanding the molecular mechanisms by which fungi interact with the other gut-residing microorganisms may enlighten approaches to rebalance and maintain the microbiome and consequently help patients to prevent flare-ups of symptoms. This is particularly relevant, as the gut microbiome, under physiological conditions, confers protective effects to the host by reinforcing the gut mucosal barrier, inhibiting pathogen colonization, and producing short-chain fatty acids (SCFAs) as a result of the fermentation of dietary fibers [22,23]. SCFAs, including propionic and butyric acid, modulate gene expression by binding to G-protein-coupled receptors, ultimately providing benefits to the health of the host [24]. Under certain conditions, such as dysbiosis, the microbiome alteration can negatively affect SCFA production [25,26], leading to detrimental side effects. For instance, Trapecar et al. [27] described a UC ex vivo model where SCFAs were able to ameliorate or worsen the colitis level based on the involvement of CD4^+^ T cells, leading to intestinal barrier disruption in the latter case.

Within PMBs, associations between fungi and bacteria, bacteria and bacteria, and fungi and fungi may be commensal, mutualistic, or antagonistic [28]. Numerous microbes have evolved to exhibit a specific attraction to neighboring species in order to survive environmental challenges [29], leading to immune system evasion, metabolic cooperation, and more efficient host colonization [30,31]. In this context, to gain insight into the mechanism(s) underlying the interactions between the two pathogenic *Candida* species when present in the same environment and how this interaction affects gut homeostasis, we evaluated the influence of *C. tropicalis* on the pathogenicity of *C. albicans.* By employing a dextran sulfate sodium (DSS)-induced colitis model in C57BL/6 (B6) mice, we assessed the susceptibility and pathogenicity in mice inoculated with only *C. albicans*, only *C. tropicalis*, or a combination of both Candida species.

We report herein that *C. tropicalis* established antagonistic interactions with *C. albicans* that affected its virulence profile, as the mice co-colonized with both *Candida* species were less susceptible to DSS-induced colitis compared to the mice inoculated with *C. albicans* only. Mechanistically, we showed that *C. tropicalis* competes with *C. albicans* for growth in the gut, decreasing the capacity of *C. albicans* to produce biofilm and, consequently, adhere to the host.

Finally, we demonstrated that the production of multiple SCFAs and the expression of genes involved in the immune response were altered when the combined *Candida* species interacted in the co-colonized mice compared to mice inoculated with only a single *Candida* species.

## 2. Materials and Methods

### 2.1. Experimental Animals

A B6 mouse colony was bred at Case Western Reserve University (Cleveland, OH, USA). The age of the mice used in the experiments was between 15 and 17 weeks. Equal numbers of males and females were used for the experiments. Micro-isolator cages (Allentown Inc., Allentown, NJ, USA) with 1/8-inch corn bedding were used to house the mice. The mice consumed laboratory rodent diet P3000 (Harlan Teklad, Indianapolis, IN, USA) during the experiments. The mice were randomized using a numerical code so that the experiments could be performed in a blinded manner. The numerical code was only revealed at the end of the experiment.

### 2.2. Fecal Pellet Homogenization

An amount of 50 g of corn bedding (including fecal pellets) was collected from each experimental cage a week before the *Candida* inoculation and blended together. The total collected pellet was then homogenized for 2 min. Then, 50 g of the homogenized pellet was redistributed to the experimental cages. This method was adopted to limit the variability between cages caused by bacterial changes.

### 2.3. Colitis Induction

Colitis was induced as described previously [32]. Briefly, mice were exposed to 3% DSS (TdB Labs AB, Uppsala, Sweden) for 7 days in drinking water to induce acute colitis. The DSS solution in drinking water was renewed every 3 days. The mice were monitored daily to assess their body weight and intestinal bleeding.

### 2.4. Yeast Strains and Growth Conditions

The *C. albicans* strain SC5314 and the *C. tropicalis* strain MRL32707 were the infecting fungi. The cells were propagated for 24 h at 37 °C in Sabouraud dextrose broth containing 50 mM glucose. The cells were then centrifuged, and the supernatant was decanted and filter-sterilized to be used for the biofilm experiments. The cell pellets were washed with phosphate-buffered saline (PBS, pH of 7.2), and standardized to 1 × 10^7^ cells/mL.

### 2.5. Biofilm Formation

Biofilms were formed as previously described [17]. Briefly, 12 mm silicone elastomer discs were cut from a sheet of silicone sheeting (Invotec International, Jacksonville, FL, USA) and used as the scaffold for handling biofilms. An amount of 4 mL of either *C. tropicalis* or *C. albicans* inoculum, standardized to 1 × 10^7^ cells/mL, was applied to each disc and left to incubate for 90 min at 37 °C (adhesion phase). The discs were then placed into new wells containing 4 mL of either 100% Sabouraud dextrose broth or 50% (*v*/*v*) Sabouraud dextrose broth/cell-free supernatant of the other organism. The discs were incubated for 24 h at 37 °C (biofilm growth phase).

### 2.6. Quantitative Measurement of Biofilms

The quantification of the yeast biofilms was performed as described previously [33]. A colorimetric method determined the mitochondrial dehydrogenase activity, an indicator of the metabolic state of the fungal cells, while a dry-weight analysis determined the total biofilm mass (fungal cells and matrix). A colorimetric assay involves the metabolic reduction of 2,3-bis-(2-methoxy-4-nitro-5-sulfophenyl)-5-[(phenylamino)carbonyl]-2H-tetrazoliumhydroxide (XTT) to a water-soluble brown formazan product whose absorbance can be read spectrophotometrically. For XTT determination, discs with biofilms were transferred to new wells containing 4 mL of PBS with 1 mg/mL of XTT and a 1 mM menadione solution. The media the biofilms were grown in was collected and centrifuged at 3500× *g* for 7 min to harvest planktonic cells for dry-mass determination. The discs containing biofilms were incubated in XTT and menadione for 5 h at 37 °C. Next, the contents of the wells containing biofilms were transferred into conical tubes and centrifuged at 3500× *g* for 7 min. A total of 1 mL of the supernatant from each tube was transferred to a cuvette and the absorbance at 520 nm was recorded using a spectrophotometer. The cell pellet was retained for determining the total dry mass of the biofilm. Each biofilm cell pellet was pooled with its respective planktonic cell pellet. The pooled suspension was filtered through a pre-weighed filter (0.45 µm pore size), air-dried at 35 °C for 24 h, and weighed. The dry mass of the biofilm and planktonic cells was determined using the mass by difference.

### 2.7. Yeast Challenge and Determination of CFUs

The *Candida* strains were first plated on Sabouraud dextrose agar and incubated at 37 °C for 2 days, and were then harvested through centrifugation. Then, a hemocytometer was used to prepare a challenge inoculum of 1 × 10^9^/mL for each strain. Each mouse was then challenged with 1 × 10^8^ blastospores in 0.1 mL of normal saline via oral gavage three times on three consecutive days (once per day, *n* = 3). This process was performed seven days before the administration of 3% DSS in the drinking water.

The quantification of colony-forming units (CFUs/mL) was determined by plating on a selective chromogenic CHROMagar culture medium [34]. The CFUs were calculated as the log CFUs per g of stools. As a control, an additional cohort of mice was challenged with the commensal yeast of the mouse gastrointestinal tract *Saccharomycopsis* (*S.*) *fibuligera* [35].

### 2.8. Histology

Colons from yeast-challenged mice were removed and fixed in 10% neutral-buffered formalin for 24 h, followed by the replacement of the formalin with 70% ethanol. Then, the tissues were embedded in paraffin and subjected to hematoxylin and eosin staining. Inflammation was assessed by a pathologist using a scoring system previously described [36]. Scores varying from 0 (normal) to 3 (maximum severity) were utilized to assess four individual histologic parameters: (1) percent ulceration, (2) chronic inflammation (macrophages and lymphocytes in the mucosal and submucosal layers), (3) percent ulceration, and (4) acute inflammation (neutrophils).

### 2.9. Colonoscopy

Colonoscopies were conducted using a flexible ureteroscope (Olympus America, Center Valley, PA, USA). Colonoscopy images were acquired on an Olympus BX41 microscope. The mice were subjected to a colonoscopy the day after the end of DSS administration. The colonic inflammation was assessed using a scoring system previously described [37]. Scores varying from 0 (normal) to 3 (maximum severity) were utilized to evaluate four individual colonoscopic parameters: (1) intestinal bleeding, (2) wall transparency, (3) perianal findings (including rectal prolapse and diarrhea), and (4) focal lesions (including ulcers and polyps). Isoflurane (Butler Schein Animal Health, Dublin, OH, USA) was utilized to anesthetize the mice prior to the colonoscopy procedure.

### 2.10. Flow Cytometry

The mesenteric lymph nodes (MLNs) were collected and crushed through a 40 μm nylon mesh. The lymphocytes were analyzed by following a previously published protocol [38]. Briefly, single-cell suspensions of MLNs were incubated with either a live/dead Fixable Violet or live/dead Fixable Blue Dead Cell Stain Kit (Thermo Scientific, Waltham, WA, USA) to determine cell viability. The cells were then washed with FACS buffer, followed by incubating them with fluorescently conjugated antibodies for 20 min at 4 °C, and then fixing them using a fixation/permeabilization buffer (eBioscience, San Diego, CA, USA) for 30 min at 4 °C in the dark. The cells were then washed with a permeabilization buffer and stained with a combination of fluorescently conjugated antibodies for 30 min at RT, to detect intracellular proteins. The following antibodies were used to detect lymphocytes expressing cytokines: an antibody cocktail with antibodies raised against CD3 (clone 145-2C11, 553067, BD Biosciences, Mississauga, ON, Canada), TNF (clone MP6-XT22, 506313, Biolegend, San Diego, CA, USA), IFNγ (clone XMG1.2, 505842, Biolegend, San Diego, CA, USA), IL-4 (clone 11B11, 3120520, Sony, Bothell, WA, USA), and IL-17 (clone TC11-18H10.1, 3134610, Sony, Bothell, WA, USA). Flow-cytometric acquisition was performed on a FACSAria sorter (BD Biosciences). The data were subsequently analyzed using the FlowJo_V10 software (Tree Star, Ashland, OR, USA). The gating strategy for lymphocytes was as follows: T lymphocytes were identified by gating on CD3^+^ live cells based on the forward vs. side scatter profiles, and then by gating on singlets using the forward scatter area vs. height, followed by dead-cell exclusion. Then, cell-subset-specific gating was performed for subpopulations positive for IFN©, IL-4, TNF, and IL-17.

### 2.11. Gas Chromatography/Mass Spectrometry Analysis

The gas chromatography/mass spectrometry (GC/MS) technique was performed to evaluate the SCFAs extracted from mouse stools through a previously described method [39]. In brief, 50 mg of stools was collected from each mouse and placed into a 1.5 mL tube containing 3.2 mm beads and 300 μL of water. The stools were then homogenized with a homogenizer (MP Biomedicals, Solon, OH, USA).

After centrifugation for 10 min at 14,000× *g*, the supernatant was placed into a new 1.5 mL tube.

A total of 100 μL of 172 mM pentafluorobenzyl bromide in acetone was then added to each tube. After incubation at 60 °C for 30 min, 250 μL of water and 500 μL of *n*-hexane were added to each tube. Next, 1 μL of each sample was inserted into the GC/MS instrument (Agilent Technologies, Santa Clara, CA, USA). Methane was utilized as the ionization gas. The ions acquired were detected in the negative mode by utilizing selected ion monitoring. Linear regression was then performed to determine the slope for each SCFA. Finally, the concentration of each SCFA was determined by using the area ratios acquired from each stool sample and the slopes previously obtained.

### 2.12. Immunohistochemistry

The colon tissues of the mice mono-inoculated with *C. albicans* or *C. tropicalis* and the co-inoculated mice were collected and fixed with 10% neutral-buffered formalin for 24 h, followed by the replacement of the formalin with 70% ethanol. Immunohistochemical (IHC) staining was performed following a previously published protocol [40]. Briefly, the samples were embedded in paraffin and then sectioned (thickness: 3–4 μm). The sections were then placed on Plus slides (Thermo Scientific, Logan, UT, USA) and deparaffinized. Then, sections were incubated in normal serum for blocking non-specific binding. H_2_O_2_ (1.75%) was utilized to block the samples for endogenous peroxidase activity. The slides were then incubated, first with a polyclonal rabbit anti-*C. albicans* primary antibody at 1:100 (PA17206; Thermo Fisher Scientific, Waltham, MA, USA) at 4 °C and then with an appropriate biotinylated goat anti-rabbit secondary antibody (Vector Laboratories, Newark, CA, USA). Next, the slides were assayed by utilizing a Vectastain ABC Kit (Vector Laboratories), and immunoreactive cells were detected using a diaminobenzidine substrate (Vector Laboratories). Finally, the slides were counterstained using hematoxylin and were then mounted by utilizing an 80% glycerol mount. Negative controls were prepared by following the same procedure in the absence of the anti-*C. albicans* primary antibody.

### 2.13. NanoString Gene Expression Analysis

Colon tissues were homogenized using 100 mg of 1.4 mm beads at 4000 rpm. Next, the total RNA was isolated using a RNeasy Mini Kit (Qiagen, Hilden, Germany). Gene expression was determined using a previously published protocol [7]. Briefly, the extracted RNA was incubated with a custom panel of 785 bar-coded probes (NanoString Technologies, Seattle, WA, USA) specific for genes associated with 50 pathways affecting the innate and immune response, the interferon response, and host susceptibility (NanoString Technologies, Seattle, WA, USA). Genes were selected from the KEGG pathways for chemokine signaling, (04062), B cell receptor signaling (04662), Th1 and Th2 cell differentiation (04658), inflammatory bowel disease (05321), and other relevant pathways. As criteria of differential expression, we used the following: *p* ≤ 0.05 and fold change >1.5. Reporter probes, the sample, capture probes, and a hybridization solution were mixed together and hybridized overnight at 65 °C. After hybridization, the samples were processed in the NanoString Prep Station, which washed away excess probes and purified the target/probe complexes through the use of magnetic beads. The target/probe complexes were then hybridized to magnetic beads complementary to sequences on the reporter probe. The purified target/probe compounds were then placed into a cartridge for data collection. A data analysis was performed using the ROSALIND^®^ online platform (https://rosalind.onramp.bio, accessed on 6 March 2024). Heatmaps and volcano plots showing the clustering of genes differently expressed among the experimental groups were obtained by utilizing the “partitioning around medoids” algorithm combined with the “Flexible Procedures for Clustering” R library and multiple database sources, including WikiPathways [41], Gene Ontology Resource [42], and NCBI [43].

### 2.14. Statistical Analysis

The flow cytometry, GC/MS, and NanoString experiments were conducted in duplicate. The CFU determination, endoscopy, histology, and XTT experiments were conducted in triplicate. The univariate and multivariate analyses were conducted using the collective data from replicated experiments. When the data fulfilled the assumptions for parametric statistics, comparisons of continuous data across experimental groups were conducted using Student’s *t*-test. Alternative nonparametric tests were used for data with unfulfilled distributional assumptions regarding the normality of the data. The data were expressed as the standard error of the means (SEMs), and 95% confidence intervals were reported when appropriate. An alpha level of 0.05 was considered significant. All analyses were conducted with GraphPad software (Version 10.0.01 (170), San Diego, CA, USA).

All authors had access to the study data and reviewed and approved the final manuscript.

## 3. Results

### 3.1. C. tropicalis Inoculation Decreases Susceptibility to DSS-Induced Colitis in C. albicans-Challenged B6 Mice

Our initial hypothesis was that an oral inoculation of *C. tropicalis* would modify the susceptibility to chemically (DSS) induced colitis in *C. albicans*-challenged mice. In order to test this hypothesis, we evaluated the severity of colitis induced by DSS administration in B6 mice pre-inoculated with *C. albicans* alone, *C. tropicalis* alone, or a combination of *C. albicans* and *C. tropicalis*. A histological analysis of the colons showed a significant decrease in the severity of colitis in the co-inoculated mice (with both *Candida* species) compared to mono-inoculated mice (only *C. albicans* challenge) (*p* < 0.05), as demonstrated by decreased cell infiltration and mucosal damage in the co-inoculated group (Figure 1A,B). No significant difference was detected between the two *Candida* mono-inoculated groups. Interestingly, the mice inoculated with only *C. tropicalis* did not exhibit significantly increased colitis compared to the co-inoculated group (*p* = ns). Furthermore, the histological results were confirmed by the endoscopic analysis, showing that the co-inoculated mice had a significantly lower percentage of intestinal ulcers and erosions and a decreased thickness of the intestinal mucosa compared to the *C. albicans*-inoculated mice (*p* < 0.05) (Figure 1C,D). A fourth group of mice was challenged with the nonpathogenic fungus (*S.*) *fibuligera* and used as a negative control to confirm whether the higher susceptibility to chemically induced colitis was specifically caused by *Candida* species. The colonoscopy and histology results confirmed that *S. fibuligera*-inoculated mice showed a significant decrease in their susceptibility to DSS-induced colitis compared to *C. albicans*- and *C. tropicalis*-challenged mice (*p* < 0.05).

### 3.2. C. tropicalis Alters the Lymphocytic Immunophenotype in C. albicans-Challenged Mice during DSS-Induced Colitis

Next, to determine if there was an immunological cause for the previously mentioned anti-inflammatory effect of a *C. tropicalis* inoculation in *C. albicans*-challenged mice, we characterized the lymphocytic immunophenotype in the MLNs of *Candida* co-inoculated vs. mono-inoculated mice at the end of the DSS treatment. Among the lymphocytes isolated from the four experimental groups, we found an increased frequency of CD3^+^ cells expressing interleukin (IL)-4, IL-17, and tumor necrosis factor (TNF) (*p <* 0.05) in *Candida* co-challenged mice compared to mice inoculated with *C. albicans* or *C. tropicalis* alone (Figure 2A–C) (Table 1). Moreover, we found an increased frequency of CD3^+^ cells expressing interferon gamma (IFN)γ in the *Candida* co-challenged group compared to the *C. tropicalis*-inoculated mice (*p <* 0.05), and the same trend was detected in relation to the single *C. albicans*-inoculated group; however, this difference did not reach statistical significance (Figure 2D). These results collectively suggest that a *C. tropicalis* inoculation renders *C. albicans*-challenged mice less susceptible to chemically induced colitis due to immune alterations associated with Th1, Th2, and Th17 lymphocytic responses.

### 3.3. C. tropicalis and C. albicans Co-Colonization Alters Production of SCFAs by Gut Microbiome

Next, to clarify the mechanism(s) by which the *C. tropicalis* and *C. albicans* co-inoculation effectively decreased acute colitis in the challenged mice compared to the mono-inoculated mice, we analyzed the metabolic basis for the anti-inflammatory effect of the *Candida* co-inoculation. We quantified SCFAs in the stools collected from the four experimental groups post-treatment. Our results showed a significant decrease in propionic acid (*p <* 0.05) (Figure 3A) and valeric acid (*p <* 0.02) (Figure 3B) in the fecal samples of *Candida* co-inoculated mice compared to the mice inoculated with *C. albicans* or *C. tropicalis* alone. Furthermore, heptanoic acid was significantly decreased in the co-challenged group compared to *C. tropicalis*-inoculated mice (*p <* 0.02), but not compared to the *C. albicans*-inoculated group (*p* = ns) (Figure 3C). Additionally, there were no significant differences between groups in relation to butyric acid and hexanoic acid (Figure 3D,E). Overall, these data indicate that the microbiome alteration caused by the *C. tropicalis* inoculation in *C. albicans*-challenged mice led to an altered production of SCFAs in comparison with mice inoculated with *C. albicans* only.

### 3.4. C. tropicalis Negatively Affects the Virulence of C. albicans by Impairing Its Ability to Produce Biofilm and Adhere to the Surface of the Host

Next, to determine if *C. tropicalis* negatively affects the virulence of *C. albicans* and its capacity to increase the susceptibility to DSS-induced colitis by impairing its ability to adhere and grow on the epithelial surface of the host, we assessed the ability of the two *Candida* species to produce biofilm in the co-inoculated vs. mono-inoculated groups. First, we measured the CFUs collected from fecal samples of the experimental groups inoculated with *C. albicans* alone, *C. tropicalis* alone, or a combination of both fungi. Our results showed that *C. albicans* CFUs (green) dramatically decreased in the group co-inoculated with both *Candida* species, compared to *C. albicans* CFUs in the group colonized by *C. albicans* only (*p <* 0.0001) and the *C. tropicalis* CFUs (purple) in the co-inoculated group (*p <* 0.02) (Figure 4A,B). Interestingly, we detected a trend related to increased *C. tropicalis* CFUs in the co-inoculated group compared to the *C. tropicalis* CFUs cultured from the experimental group infected with *C. tropicalis* alone. To further assess if the major cause of decreased *C. albicans* virulence was due to an impaired adherence, we performed an IHC analysis to detect the presence and the location of *C. albicans* in the epithelium of mice mono-inoculated with *C. albicans* compared to co-inoculated mice. Our results unequivocally showed that *C. albicans* was less present in the co-inoculated group compared to the mice inoculated with *C. albicans* only. Moreover, *C. albicans* was able to penetrate deeper into the mucosal layer when present as the sole yeast, compared to the co-inoculated group (Figure 4C). Colons from the group inoculated with *C. tropicalis* alone stained with *C. albicans* antibodies showed no clearly positive stain, demonstrating the specificity of the utilized primary antibody for the *C. albicans* species.

Finally, we examined the capacity of *C. tropicalis*-produced metabolites to decrease the biofilm production of *C. albicans* in vitro using a dry-weight assay and a colorimetric XTT assay. The dry-weight results showed that the medium collected from *C. tropicalis* cultures was effective in reducing the ability of *C. albicans* to produce biofilm (*p <* 0.05) (Figure 4D). Interestingly, the supernatant collected from the *C. albicans* culture failed to decrease the biofilm production activity when added to the *C. tropicalis* strain (*p* = ns). Furthermore, the XTT assay indicated a trend towards decreased biofilm production for both *C. tropicalis* incubated with the medium collected from the *C. albicans* culture and *C. albicans* incubated with the medium collected from *C. tropicalis*, although not to a statistically significant level (Figure 4E). These data clearly indicate that *C. tropicalis* competes with *C. albicans* for growth in the same host, limiting the capacity of *C. albicans* to adhere to the host intestinal surface, affecting the microbiome differently, and consequently decreasing the susceptibility to DSS-induced colitis in challenged mice.

### 3.5. C. tropicalis Altered the Expression of Genes Involved in Multiple Immune Responses in C. albicans-Challenged Mice

In order to analyze the potential mechanism(s) associated with a decreased level of colitis and possible immunologic alterations in the mice inoculated with the combined *Candida* species, we used a NanoString “host response” panel targeting 785 genes involved in the innate and adaptive immune response, the interferon response, and host susceptibility (NanoString Technologies, Seattle, WA, USA). The acquired data are shown in the heatmap graphs (Figure 5A,B), which highlight the clusters of significantly altered genes in the mice co-inoculated with both *Candida* species in comparison with the mice mono-inoculated with *C. tropicalis* or *C. albicans*. Specifically, in the co-inoculated mice, nine genes were significantly upregulated compared to the mice inoculated with *C. tropicalis* alone, and the altered imputed pathways included IL-1 and IL-6 signaling, a complement cascade, and leukocyte chemotaxis, including *Il10*, *Cxcl2*, *Cxcl14*, *Ccl4*, and *Il1rn* (*p <* 0.05) (Figure 5C); moreover, the co-inoculated mice had two upregulated genes and one downregulated gene compared to the mice inoculated with *C. albicans* alone. The involved altered pathways included a complement cascade and interferon signaling, including *Ccr9*, *Agt*, and *Nkg7* (*p <* 0.05) (Figure 5D).

These results strongly indicate that *C. tropicalis* and *C. albicans* interactively compete in the same host niche, leading to changes in the expression of genes involved in multiple immune response pathways and eventually resulting in a decreased susceptibility to colitis in the combined condition.

## 4. Discussion

We report that an inoculation with *C. tropicalis* is associated with a decreased severity of DSS-induced colitis in B6 mice challenged with *C. albicans*. The decreased pathogenicity of the co-inoculated mice was indicated by significantly reduced histological and colonoscopic scores, and resulted in a better physiological outcome. The same trend was detected in relation to the *C. tropicalis* mono-inoculated group, although the difference was not statistically significant. This observation suggests that the described antagonistic interaction between the two *Candida* species may be bi-directional, where *C. albicans* also produces metabolites capable, at some level, of affecting the metabolic activity of *C. tropicalis*.

In the present work, we also defined the immunological aspects of the decreased colitis that occurred in the experimental group inoculated with *C. albicans* and *C. tropicalis combined.* First, we detected the upregulation of Th-2 immunity in the co-inoculated mice with a significantly increased IL-4 production compared to the mice challenged with *C. albicans* alone. Our data agree with those of Mencacci et al. [44], who found that endogenous IL-4 is essential for stimulating CD4^+^ defensive responses against *C. albicans* through the activation of the adaptive and innate immune systems. Furthermore, the levels of IL-4 measured in the co-inoculated group were similar to those of the control group. Interestingly, these data are consistent with our previous study [45], which showed significantly decreased *Il4* expression in mice inoculated with a single *Candida* species compared to the control group. Second, we also detected a significant downregulation of Th-17 immunity in the mice challenged with *C. albicans* compared to the control group. These data are expected, given the well-known protective role of Th-17 cells in antifungal immune responses, where several genetic anomalies involving the IL-17 signaling cascade have been proven to increase the susceptibility to mucocutaneous candidiasis in multiple mouse models and human subjects [46,47]. Conversely, the level of IL-17 produced in the co-inoculated mice was significantly higher compared to the mice inoculated with *C. albicans* only. This can be explained by the interesting theory that *C. tropicalis* and *C. albicans*, rather than having a mutualistic interaction, compete for adherence and growth in the same host niche, affecting the microbiome and the immune response differently compared to when they are present as the sole fungal source. This theory is supported by the observation that several microorganisms present in the gut have different effects based on their interaction with other microbial species. For example, published data have shown that *C. albicans* has a protective effect in DSS-treated germ-free mice as well as in antibiotic-treated specific-pathogen-free mice, two models used to study the effect of *Candida* species in the context of a depleted/absent gut microbial community [48]. By contrast, another study published by Jawhara et al. [49] showed, in accordance with our results, that the decrease in *C. albicans* caused by a *Saccharomyces boulardii* inoculation was positively correlated with a decreased susceptibility to DSS-induced colitis. Our data suggest that the specific bacterial populations responsible for the increased susceptibility to chemically induced colitis are those microbes also affected by a *C. albicans* inoculation, thus explaining the divergence in the outcome between our own studies with those published previously. Our theory is further corroborated by the GC/MS results, indicating an alteration in certain SCFAs as the likely result of the close interactions between yeasts and bacteria that take place during chemically induced colitis. Specifically, propionic and valeric acid were found to be consistently decreased in the stool samples of co-inoculated mice compared to those from *Candida* mono-inoculated mice. Bacterial-produced propionic acid has been generally shown to provide beneficial effects to the host [27], but, as stated in the introduction, under particular conditions such as dysbiosis, SCFAs can lead to detrimental side effects. For instance, previous in vitro studies have demonstrated a positive correlation between elevated levels of propionic acid and higher virulence levels of *E. coli* (specifically, its ability to penetrate and colonize phagocytes) isolated from CD patients [50]. In addition, the observed levels of valeric acid were significantly lower in the stools of the co-inoculated mice compared to those of the mice challenged with *C. albicans* only, suggesting a correlation between the dysbiosis obtained with a *C. albicans* mono-inoculation and the growth of valeric-acid-producing bacteria. Valeric acid was shown to increase the inflammatory response mediated by IL-17 signaling in a recent study [51].

Moreover, the CFU analyses showed that the quantitative recovery of *C. albicans* from the fecal samples of the experimental group inoculated with both yeasts was significantly inferior compared to the colonies obtained from the fecal material of the mice inoculated with *C. albicans* alone. This observation strongly suggests that *C. tropicalis* was able to reduce the growth and adherence capacity of *C. albicans*. To corroborate our hypothesis, the in vitro dry-weight assay highlighted a marked decrease in *C. albicans* biofilm production for *C. albicans* cultured in the presence of *C. tropicalis* supernatant, demonstrating that certain metabolites produced by *C. tropicalis* impair the biofilm-forming capacity of *C. albicans*. Our data are in agreement with those of Santos et al. [52], who demonstrated that *C. tropicalis* was capable of limiting *C. albicans*’ metabolic activity and its capacity to form colonies in mixed biofilms. It should also be noted that the 50:50 media ratio utilized (50% Sabouraud dextrose broth and 50% cell-free supernatant of the other organism) implies that a minor quantity of glucose and other nutrients was available to the *Candida* species for biofilm formation compared to the controls (*Candida* species incubated with 100% fresh media); therefore, the comparison between the results may reveal the impact not just of the secreted molecules, but also of the nutrient limitation. However, we do not believe that the effect observed was caused by a nutrient limitation, since *C. tropicalis*, in contrast to *C. albicans*, did not show significant differences compared to its 100% fresh-medium-incubated control based on the dry-weight assay results. Conversely, the XTT assay indicated no statistical differences between the *C. albicans* incubated with a 50:50 media ratio compared to the controls. This can be explained by the XTT assay strictly measuring the metabolic activity of the adherent cells that comprised the biofilm. The dry-weight assay measured the total biomass of the adherent cells in the biofilm, as well as the planktonic cells that propagated from the biofilm as it matured. The adherent cells that comprised the *C. albicans* biofilms incubated with *C. tropicalis* supernatant had a similar biomass compared to the controls, hence why the XTT results did not show differences between these biofilms. However, the planktonic biomass of the *C. albicans* biofilms incubated with *C. tropicalis* supernatant was lower than the planktonic biomass of the controls. This is why the total mass of the biofilms differed; the difference lies in the decreased amount of planktonic cell growth relative to the control. This difference was not captured by the XTT assay.

The quantification of *C. tropicalis* CFUs in the co-inoculated mice was similar to the level of CFUs quantified in the group inoculated with *C. tropicalis* alone, and in accordance with these data, the dry-weight assay showed that the *C. tropicalis* biofilm production was not significantly altered in the *C. tropicalis* cultured with *C. albicans* supernatant. Our hypothesis was further supported by IHC staining specific for *C. albicans*, showing that this yeast was not only more abundant, but it was also able to penetrate deeper into the epithelium and the lamina propria of the mice mono-inoculated with *C. albicans* compared to the co-inoculated group, where *C. albicans* was mainly located on the epithelial surface.

Lastly, the NanoString analysis showed that the *C. tropicalis* inoculation not only drastically limited the virulence and the growth of *C. albicans*, but it also critically affected the expression of three genes implicated in the complement cascade and interferon signaling in the *C. albicans*-challenged mice, via genes such as *Ccr9*, *Agt*, and *Nkg7*. These results are corroborated by multiple studies indicating how various polymicrobial interactions [53] and adaptive and innate immune responses [54] can ameliorate or worsen IBD symptoms. In particular, our data are in agreement with a study by Wurbel et al. [55], which highlighted a strong correlation between *Ccr9* expression and the amelioration of DSS-induced colitis symptoms. Specifically, their results showed that CCR9-knockout mice were more susceptible to DSS-induced colitis compared to wild-type controls, and that a dysregulated Th-17 immune response involving different macrophage subsets was observed during their recovery period following DSS treatment.

## 5. Conclusions

In conclusion, these data strongly indicate an antagonistic microbial interaction between the two *Candida* species *C. albicans* and *C. tropicalis*, where *C. tropicalis* may produce molecules capable of limiting the capacity of *C. albicans* to adhere to the host intestinal surface, form polymicrobial associations, and, consequently, negatively affect virulence factors, thus making the combined inoculation less harmful in DSS-treated mice. The fungal competitive interaction highlighted in this study may explain the reason why the incidence of invasive candidiasis in immunocompromised patients characterized by the detection of multiple *Candida* species is less than 10% compared to the incidence of candidiasis characterized by the detection of a single yeast [56].

This is the first study attempting to clarify the interactions between *C. tropicalis* and *C. albicans* in the context of chemically induced (DSS) colitis. However, the exact molecular mechanisms involved in the interactions between these two fungal species need to be further investigated in order to unequivocally identify the metabolic pathways associated with the described antagonistic effect and to discover novel molecules that can alter the pathogenicity of *C. albicans*.

## Figures and Tables

**Figure 1 jof-10-00245-f001:**
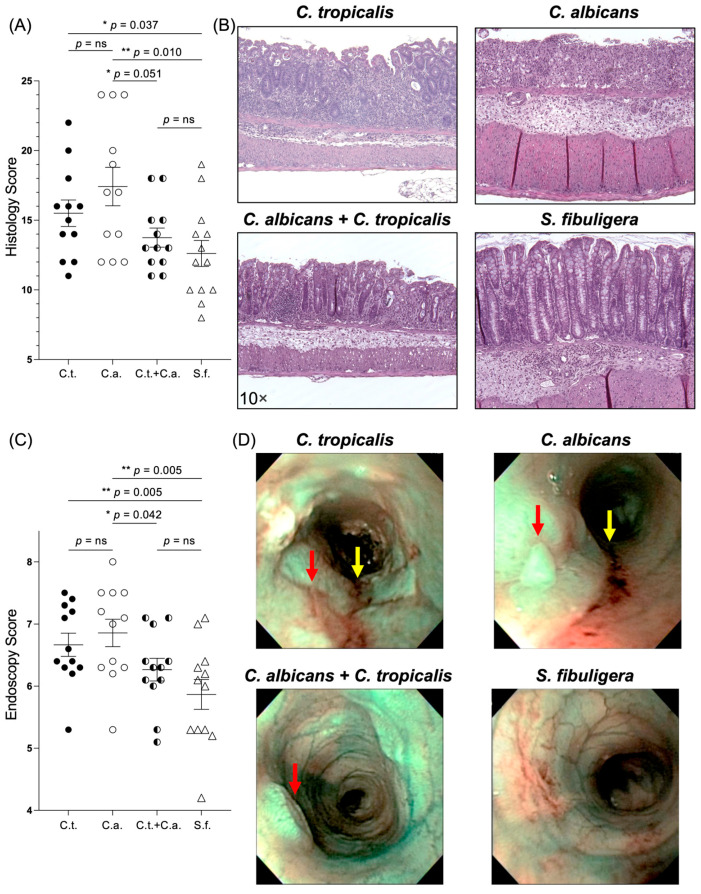
*Candida (C.) tropicalis* inoculation decreased susceptibility to chemical-induced colitis in *C. albicans*-challenged C57BL/6 mice. (**A**) Histological analysis showed decreased colonic inflammation in co-inoculated mice compared to mice challenged with *C. albicans* alone (unpaired *t*-test, 13.75 ± 0.69 vs. 17.50 ± 1.36; *p* < 0.05; N = 12/group) and increased colonic inflammation in *C. albicans*-challenged mice and *C. tropicalis*-challenged mice compared to control group (17.50 ± 1.36 vs. 12.62 ± 0.93; *p* < 0.02) (15.50 ± 0.95 vs. 12.62 ± 0.93; *p* < 0.05). No statistical differences were found between co-inoculated mice and control group (13.75 ± 0.69 vs. 12.62 ± 0.93; *p* = ns) or between co-inoculated mice and *C. tropicalis*-inoculated group (13.75 ± 0.69 vs. 15.50 ± 0.95; *p* = ns). (**B**) Representative colonic histopathological sections of *C. albicans*- and *C. tropicalis*-inoculated mice show the presence of ulcers, active cryptitis, increased inflammatory cells in the lamina propria, and thicker intestinal mucosa compared to co-inoculated mice and control group, showing minimal inflammatory cells and mild active cryptitis. (**C**) Colonoscopic evaluation showed increased colitis in distal colon of mice challenged with *C. albicans* alone compared to co-inoculated mice (6.86 ± 0.22 vs. 6.27 ± 0.18; *p <* 0.05) and control group (6.86 ± 0.22 vs. 5.87 ± 0.24; *p <* 0.02). No statistical differences were found between co-inoculated mice and control group (6.27 ± 0.18 vs. 5.86 ± 0.24; *p* = ns) or between co-inoculated mice and *C. tropicalis*-inoculated group (6.27 ± 0.18 vs. 6.67 ± 0.19; *p* = ns). (**D**) Narrow-band imaging endoscopic pictures of distal colon showed higher presence of ulcers (red arrows) and colorectal bleeding (yellow arrows) in *C. albicans*-inoculated and *C. tropicalis*-inoculated mice compared to co-inoculated mice and control group. Data are expressed as mean ± SEM; * *p <* 0.05, ** *p <* 0.02.

**Figure 2 jof-10-00245-f002:**
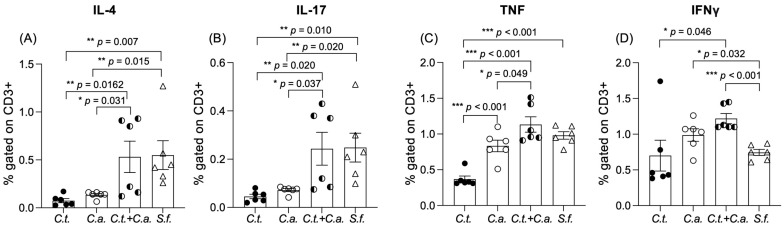
*C. tropicalis* altered lymphocytic immunophenotype in *C. albicans*-challenged mice during dextran-sodium-sulfate-induced colitis. Cultured lymphocytes collected from mesenteric lymph nodes displayed significant differences between *C. albicans*-inoculated mice, *C. tropicalis*-inoculated mice, co-inoculated mice, and control group: (**A**) decrease in interleukin (IL)-4 between *C. albicans*-inoculated mice and co-inoculated mice; decrease in IL-4 between *C. tropicalis*-inoculated mice and both co-inoculated mice and control group. (**B**) Decrease in IL-17 between *C. tropicalis*- and *C. albicans*-inoculated mice compared to co-inoculated mice and control group. Lymphocytes did not display significant difference between IL-4 and IL-17 in co-inoculated mice versus control group. (**C**) Decrease in tumor necrosis factor (TNF) between *C. albicans*-inoculated mice and co-inoculated mice; decrease in TNF between *C. tropicalis*-inoculated mice and co-inoculated mice, and between *C. albicans*-inoculated mice and control group. (**D**) Decrease in interferon (IFN)γ between *C. tropicalis*-inoculated mice and co-inoculated mice; increase in IFNγ between *C. albicans*-inoculated mice and control group; and increase in IFNγ between co-inoculated mice and control group. Data are expressed as mean ± SEM; * *p <* 0.05, ** *p <* 0.02, *** *p <* 0.001.

**Figure 3 jof-10-00245-f003:**
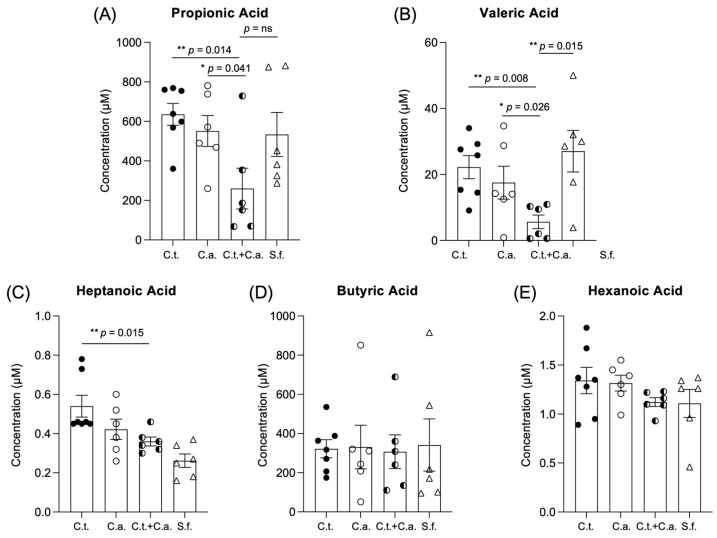
*C. tropicalis* and *C. albicans* co-colonization alters production of short-chain fatty acids by gut microbiome. Gas chromatography/mass spectrometry (GC/MS) analysis indicates decreased levels of the following: (**A**) propionic acid in fecal samples of co-inoculated mice compared to mice challenged with *C. albicans* alone (unpaired *t*-test: 259.6 ± 103.1 vs. 551.4 ± 78.18; *p <* 0.05) or *C. tropicalis* only (259.6 ± 103.1 vs. 635.4 ± 55.45; *p <* 0.02); no differences were found between co-inoculated mice and control group (259.6 ± 103.1 vs. 533.8 ± 111.4; *p* = ns). (**B**) Valeric acid in co-inoculated mice compared to mice challenged with *C. albicans* alone (5.653 ± 2.072 vs. 17.520 ± 4.983; *p <* 0.05), *C. tropicalis* alone (5.653 ± 2.072 vs. 22.230 ± 3.480; *p <* 0.02), or control group (5.653 ± 2.072 vs. 27.060 ± 6.288; *p <* 0.02). (**C**) Heptanoic acid in co-inoculated mice compared to mice challenged with *C. tropicalis* alone (0.360 ± 0.023 vs. 0.540 ± 0.056; *p <* 0.02). No differences were found between groups in terms of (**D**) butyric acid or (**E**) hexanoic acid (*p* = ns). Data are expressed as mean ± SEM; *N* ≥ 6/group; * *p <* 0.05, ** *p <* 0.02.

**Figure 4 jof-10-00245-f004:**
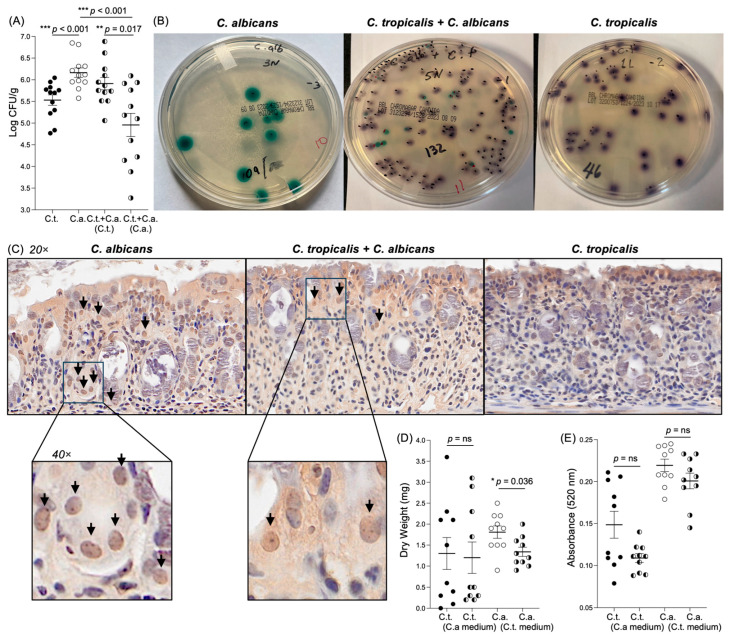
*C. tropicalis* negatively affects the virulence of *C. albicans* by impairing its ability to produce biofilm and adhere to the surface of the host. (**A**) The recovery of *C*. *albicans* and *C. tropicalis* from fecal samples of the co-inoculated mice and mono-inoculated mice. Shown are colony-forming unit (CFU) counts from weighed, homogenized, and plated stools on CHROMagar. CFU assays showed that the *C*. *albicans* burden significantly decreased in co-inoculated mice compared to mice inoculated with *C. albicans* alone (unpaired *t*-test: 4.958 ± 0.267 vs. 6.164 ± 0.109; *p <* 0.001) and compared to *C. tropicalis* CFUs in the co-inoculated group (4.958 ± 0.267 vs. 5.912 ± 0.144; *p <* 0.02). (**B**) Representative pictures of the *C*. *albicans* and *C. tropicalis* recovery 24 h after the last inoculum. (**C**) Immunohistochemical staining for *C. albicans* showed that *C. albicans* was more abundant and was able to penetrate deeper into the epithelium and the lamina propria (black arrows) of the colon tissues collected from the mice mono-inoculated with *C. albicans* compared to the co-inoculated group. The colon tissues of mice inoculated with *C. tropicalis* presented no clearly positive stain. Panels: 20× and 40× magnification. (**D**) The dry-weight assay showed that *C. albicans* treated with the *C. tropicalis* culture supernatant produced less biofilm compared to untreated *C. albicans* (1.340 ± 0.111 vs. 1.810 ± 0.147; *p <* 0.05), while the *C. tropicalis* strain treated with the *C. albicans* culture supernatant did not show any significant alteration related to biofilm production compared to untreated *C. tropicalis* (1.200 ± 0.374 vs. 1.300 ± 0.381; *p* = ns). (**E**) The (2-methoxy-4-nitro-5-sulfophenyl)-5-[(phenylamino)carbonyl] 2H-tetrazoliumhydroxide (XTT) assay results showed no significant differences between *C. albicans* treated with the *C. tropicalis* culture supernatant and untreated *C. albicans* (*p* = ns) or between the *C. tropicalis* strain treated with the *C. albicans* culture supernatant and untreated *C. tropicalis* (*p* = ns). The data are expressed as the mean ± SEM; N = 10/group; * *p <* 0.05, ** *p <* 0.02, *** *p <* 0.001.

**Figure 5 jof-10-00245-f005:**
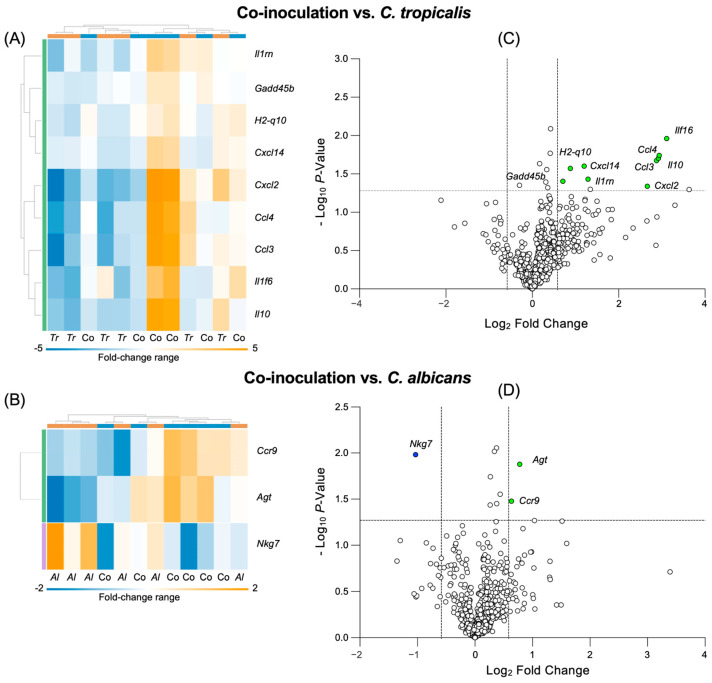
*C. tropicalis* altered the expression of genes involved in multiple immune responses in *C. albicans*-challenged mice. Heatmaps of the normalized data indicated connections between genes differentially expressed in the colonic tissues of the co-inoculated (Co) mice compared to the mice mono-inoculated with (**A**) *C. tropicalis* (*Tr*) or (**B**) *C. albicans* (*Al*). Data are shown indicating associations between the gene expression (gold, upregulation; blue, downregulation) and treatment. Each row corresponds to a specific probe and each column corresponds to a specific sample. Hierarchical clustering was used to generate dendrograms. Volcano plots expressing the NanoString data for 785 genes showed that the colonic tissues of the co-inoculated mice had (**C**) nine genes upregulated compared to the mice inoculated with *C. tropicalis* alone and (**D**) one gene downregulated and two genes upregulated compared to the mice inoculated with *C. albicans* alone. Fold-change > ±1.5, *p <* 0.05.

**Table 1 jof-10-00245-t001:** A comparison of the cytokines produced by CD3^+^ MLN lymphocytes among the *Candida*-challenged groups. An unpaired *t*-test highlighted the statistical differences based on the frequency of CD3^+^ cells expressing IL-4, IL-17, TNF, and IFNγ among the *C. albicans*, *C. tropicalis*, and *S. fibuligera* (control) mono-inoculated mice and *Candida* co-inoculated mice. * *p <* 0.05, ** *p <* 0.02, *** *p <* 0.001.

Cytokines	*C.t.* vs. *C.a.*	*C.t.* vs. (*C.t.* + *C.a.*)	*C.a.* vs. (*C.t.* + *C.a.*)	*C.t.* vs. *S.f.*	*C.a.* vs. *S.f.*	(*C.t.* + *C.a.*) vs. *S.f.*
IL-4	*p* = ns	** *p* = 0.016	* *p* = 0.031	** *p* = 0.007	** *p* = 0.015	*p* = ns
IL-17	*p* = ns	** *p* = 0.020	* *p* = 0.037	** *p* = 0.010	** *p* = 0.020	*p* = ns
TNF	*** *p* < 0.001	*** *p* < 0.001	* *p* = 0.049	*** *p* < 0.001	*p* = ns	*p* = ns
IFNγ	*p* = ns	* *p* = 0.046	*p* = ns	*p* = ns	* *p* = 0.032	*** *p* < 0.001

## Data Availability

The data that support the findings of this study are available from the corresponding author, L.D.M., upon reasonable request. The data will be stored for a long-term period (a minimum of 5 years) in the Box storage service (hosted in the cloud) that enables Case Western Reserve University to store, access, and share files securely. Box is the only approved platform for storing restricted data in the cloud at Case Western Reserve University.

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
