# Peer review of "Candida tropicalis Affects Candida albicans Virulence by Limiting Its Capacity to Adhere to the Host Intestinal Surface, Leading to Decreased Susceptibility to Colitis in Mice"

_jof, 2024, doi:10.3390/jof10040245_

Round 1

Reviewer 1 Report

The authors investigate the interaction between Candida tropicalis and Candida albicans in the context of colitis in mice. This is a novel study indicating substantial interaction between these two yeasts. The authors state that C. tropicalis negatively influences the virulence of C. albicans, although the interaction may be bi-directional, this is not mentioned clearly and should perhaps be included in the discussion of the results where appropriate.

Other comments:

Materials and Methods

1. Please provide references for the various methods used

2. For the methods used for biofilm formation (ln 129-136), when the cell free supernatant of the one organism is tested in a 50:50 ration to media, there would be less glucose (and other nutrients) available to the organisms for biofilm formation than in fresh media. The comparison between these results may therefore not just reveal the impact of secreted molecules in the supernatant, but also nutrient limitation.

3. The description of the flow cytometry methods should be improved. It is unclear which antibodies were used.

4. Similarly, the samples used for immunohistochemistry are not described in the materials and methods

Results

5. For the results presented in figure 1 A and 1C and discussed, the authors should indicate if there was any significant difference between C. albicans and C. tropicalis monospecies infections. If these were not significantly different, then co-infection may not only influence C. albicans virulence, but also that of C. tropicalis.

6. In the materials and methods the authors state that they quantified biofilms using both XTT assays as well as biomass. However, in the results (ln 390) they only mention that this was done using XTT assays, but in figure 4D, the Y-axis gives biomass (mg). Please provide and discuss the data for both XTT and biomass determination as these are two very different measures of biofilms.

7. The recovery of yeasts from fecal samples was performed on SDA (ln 160). How was this medium able to distinguish between C. albicans and C. tropicalis? The authors show different coloured colonies for the two species, but this is not possible on normal SDA.

8. The data presented in figure 5 A,B, C do not group the two conditions together on the X-axis. Thus it seems as if no clear pattern of gene regulation between the co-inoculated and the various mono species is present. Please explain this and the conclusions drawn from this data.

Minor comment:

ln 146: scrapped should be replaced with scraped

These are given above

Reviewer 2 Report

Is there evidence for co-infections/have co-infections been reported in the literature? If so, please give examples. If not, please provide a rationale for it.

Can the authors please provide detailed information on how mice were infected? What specifically is meant by “inoculated”. In addition, when Candida strains were grown for 2 days at 37 to prepare the inoculum, how accurate are the cell counts? What was the percentage of yeast to filamentous growth?

The description of the statistical analyses must be expanded. Please specify which experiments were done in duplicates. XTT assays should be done at least in triplicates. In addition, please specify the exact tests done in Prism and for which data. What exactly is meant by continuous data?

Is Saccharomycopsis fibuligera a commensal of the GI tract? This should be stated and referenced to support the choice of control especially because it is used as control for the statistical analyses. It might be better to use single species results as controls for the co-infection experiments.

Figure 2: How were the percentages for each cytokine determined? Is that compared to the total number of cells? It is not clear to me. The figure legend is quite excessive and hard to follow/understand. I recommend having a separate table with the t-test results and make the legend more concise that way.

Figure 3: what is the connection of SFCA levels and mechanism of decreasing infection? This should be supported by references and a brief explanation. Also, I am not sure how the authors can say that there was a “significant bacterial change”. Are they using SFCA values as a proxy for bacterial interactions? I strongly suggest to add some content to the introduction (see my comment further below on discussion).

Figure 4A: What is the growth rate in vitro for single species and co-inoculated species for media used to prepare the in vivo inoculum? If C. tropicalis is growing slower in vitro that may contribute to the significant difference seen in vivo.

Figure 4C: I am surprised that there are no hyphal cells. How do the authors think the deeper penetrations might occur? And why are there no hyphal cells? Also, it may have been informative to use fluorescently tagged strains to examine the tissue penetration.

Figure 5: How relevant is the comparison of co-infected versus the control? The legend for the fold change range needs to be below the strain names and it might be better to show them vertically to the right or left of the heat map. In addition, the sample letters are not informative enough and need to be improved and/or defined. What do the red boxes mean?

The fold change (I think this is the log 2 fold change?) cut off is quite low (> 1.25). Commonly, a fold change of 2 is used. I suggest to re-plot the volcano plots using a Log2 cutoff of 1 (2 fold).

I also would be careful to draw conclusions about altered pathways just from gene expression data. It would be better to look at GO terms or do a pathway analysis.

Have the authors thought about doing adherence and endocytosis ex vivo experiments to support the in vivo findings?

Have the authors looked at the micro architecture of the single-species and double-species biofilms?

Discussion: Because the authors did not look at interactions between Candida species and bacteria, they cannot claim any association unless they explain in more detail the connection between SFCA and bacteria. Specifically, the explanation on lines 469-478 really needs to go into the introduction, and then can be revisited in the discussion.

Minor:

Lines 234/235: Sentence structure

Line 429: Please specify what the other 3 experimental groups are.

Round 2

Reviewer 1 Report

The manuscript has been improved as suggested

The manuscript has been improved as suggested